# Electrodermal activity patient simulator

Gregor Geršak⊙*⊙, Janko Drnovšek⊙

University of Ljubljana, Faculty of Electrical Engineering, Ljubljana, Slovenia

⊙ These authors contributed equally to this work.
* gregor.gersak@fe.uni-lj.si

## Abstract

Electrodermal activity (EDA) is an electrical property of the human skin, correlated with person's psychological arousal. Nowadays, different types of EDA measuring devices are used in highly versatile fields–from research, health-care and education to entertainment industry. But despite their universal use the quality of their measuring function (their accuracy) is questioned or investigated very seldom. In this paper, we propose a concept of an EDA patient simulator—a device enabling metrological testing of EDA devices by means of a variable resistance. EDA simulator was designed based on a programmable light-controlled resistor with a wide resistance range, capable of simulating skin conductance levels (SCL) and responses (SCR) and was equipped with an artificial hand. The hand included electrically conductive fingers for attachment of EDA device electrodes. A minimal set of tests for evaluating an EDA device was identified, the simulator's functionality discussed and some testing results presented.

**Data Availability Statement:** All data files are available from the Figshare database (DOI no. 10.6084/m9.figshare.11551224).

**Funding:** The authors acknowledge the financial support from the Slovenian Research Agency (research core funding No. P2-0225). http://www.

## Introduction

Electrodermal activity (EDA) is an electrical property of the human skin dependent on changes of sympathetic part of the autonomic nervous system. EDA of a person changes when hers/his level of arousal changes [1]. Nowadays it is used with increasing regularity, because industry and research institutions are interested in acquiring objective information of human perception of products, services or even human arousal or emotions identification during cognitive and mental tasks. The reason of EDA's popularity is relative low cost of the measuring devices, simplicity of their manufacturing and use, combined with relatively fast physiological response.

In the most simplified version, electrodermal activity is a contact non-invasive measure of human sweating, which could be a result of body thermal regulation processes or of a certain psychological arousal level of the person. Nowadays, the most common assumption is that when a person is psychologically aroused, excited or activated, hers/his EDA signal increases [1–3], although there are also other opinions [4].

EDA can be monitored within controlled laboratory environments in static, sedentary position (e.g. sitting at a computer) in controlled environmental conditions (e.g. air humidity and temperature, vibration, noise, lightning) with correspondingly less disturbances, measuring errors and unwanted moving artefacts. On the other hand, monitoring of EDA in real-life

arrs.si/en/index.asp. The funders had no role in study design, data collection and analysis, decision to publish, or preparation of the manuscript.

**Competing interests:** The authors have declared that no competing interests exist.

conditions outside laboratory provides a more ecologically valid setting, but is much more burdened by environmental conditions, moving artefacts, dynamic errors etc. [5,6].

In clinical settings and applied psychology EDA is often used for stress, pain and sleep studies [7–10]. It was used in studies of clinical conditions like schizophrenia, panic disorder, anxiety, multiple sclerosis, attention-deficit hyperactivity-disorder (ADHD), autism, Alzheimer [11–19]. EDA was used also in ICT (information and communications technology) and entertainment [20–24], education [25,26] and food industry research [27,28].

In general, biomedical devices are clinically validated by means of a comparison with a reference device, both used on an adequately large population of human subjects. Involvement of a large group of needed subjects and logistical, ethical and practical issues are the reason for complexity and high cost of clinical validations. Therefore, other means of evaluation of devices are envisaged, e.g. simplifications of comparison protocols, reduction of number of participants. One of the possible solutions are patient simulators.

Patient simulators are devices substituting patients. They offer a controlled way of evaluating biomedical measuring devices. A simulator is a device for imitating a physical phenomenon and is used for testing reliability, robustness and accuracy of a measuring device. Patient simulators are devices used for testing and/or calibrating biomedical devices, which measure physiological parameters. Their primary function is to generate signals, equal or similar to real physiological ones in order to test the measuring accuracy of devices measuring these signals. A well-known example are blood pressure (BP) patient simulators [29–31]. BP simulators are electromechanical devices, capable of generating air pressure pulses, which are fed to BP monitor. The air pulses, which can have artificial or physiological shapes are used to test the repeatability, stability and even accuracy of non-invasive BP monitors. Similarly, patient simulators for ECG, EEG, heart-rate and pulse oximetry are used [32,33].

An EDA simulator is a device capable of generating typical skin conductance (SC) waveforms, which facilitate metrological checks of any EDA measuring device in both static and dynamic conditions. As always the case with any device equipped with measuring function, also EDA devices are of different levels of metrological quality. I.e. their measuring error and measuring uncertainty can be very diverse, and consequently their reliability more or less questionable. In order to get reliable, accurate and repeatable measuring results, metrological checks or tests of EDA devices should be performed regularly.

## Electrodermal activity

EDA signal primarily contains two pieces of information–the level of the signal and the response of the signal. Tonic, slowly changing part of the SC signal is named skin conductance level (SCL). Fast phasic pulses are called electrodermal responses, or skin conductance responses (SCR). SCL value indicates the level of psychological arousal of the subject, while the number of SCRs are a measure of subject's momentary arousal and represents pulses in skin conductance signal (Fig 1). SCR occur where EDA amplitudes exceed a certain threshold in a certain time period (e.g. pulses occurring less than 9 seconds after the beginning of the increase and having amplitudes larger than 0.02 uS) [3,34]. Commonly SCRs are estimated after 0.05 Hz high pass filtering and employment of a response threshold of 0.01 to 0.05 uS [35]. Number of SCR per minute is a measure of the subject's arousal. As a rule-of-a-thumb, values of a couple of SCR per minute indicate the subject is in relaxed state (baseline) and values above 20 SCR/min indicate an aroused subject [1,34].

Typical values of raw EDA depend strongly on individuals and experimental situations and can vary considerably, but usually skin conductance level (SCL) ranges up to a couple of tens

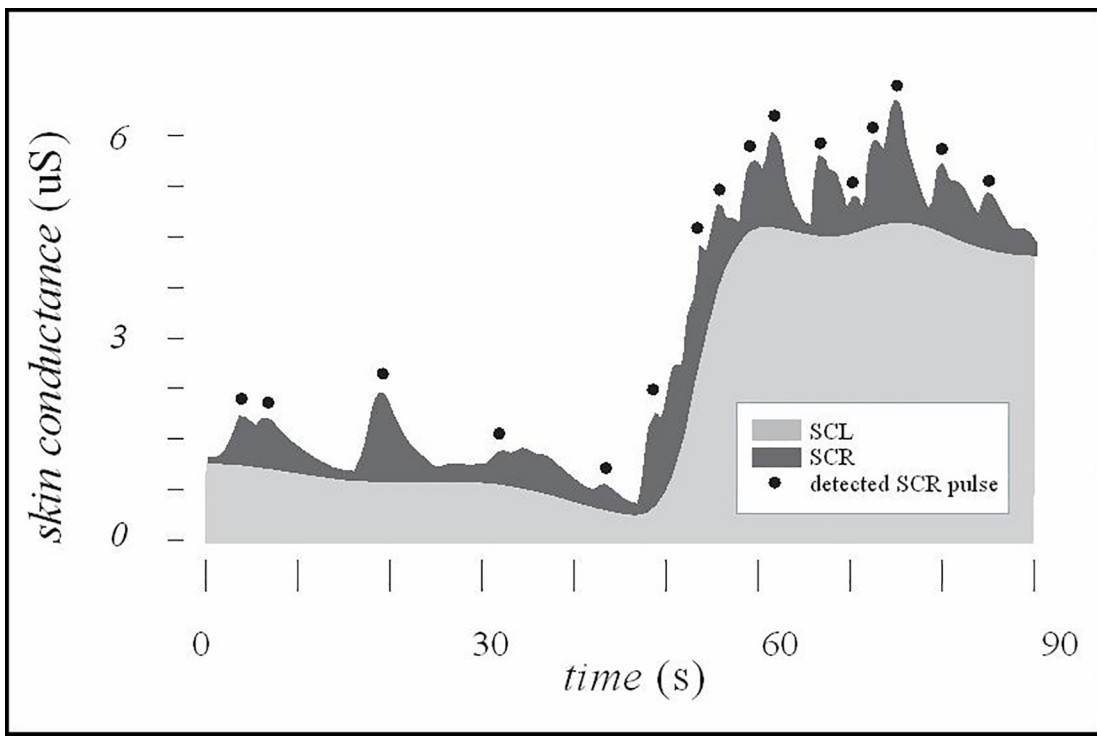

**Fig 1. Electrodermal activity with detected skin conductance responses (SCR) (marked with black dots).**

of microsiemens [1,3]. In terms of phasic skin conductance, SCR amplitudes can typically range from the threshold to a maximum of around a couple of uS.

## EDA devices

There are two general forms of EDA devices, based on two methods—endosomatic and exosomatic. Endosomatic method does not apply any external current, and exosomatic applies an external current to the skin. Three main measuring methods can be identified: i) endosomatic method, ii) AC exosomatic method (applying AC current) and iii) DC exosomatic method (applying DC current via electrodes) [1].

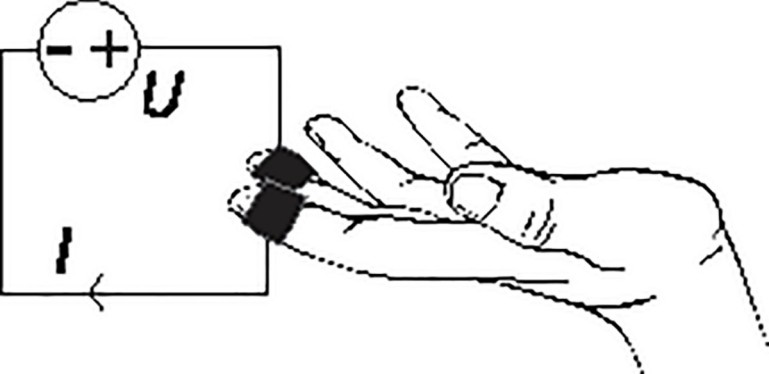

**Fig 2. Via electrodes (black) DC exosomatic measuring instrument applies a DC voltage of up to U = 1 V to the skin.** By measuring the ratio of applied voltage U and resulting current I skin conductance G can be calculated (G = I / U).

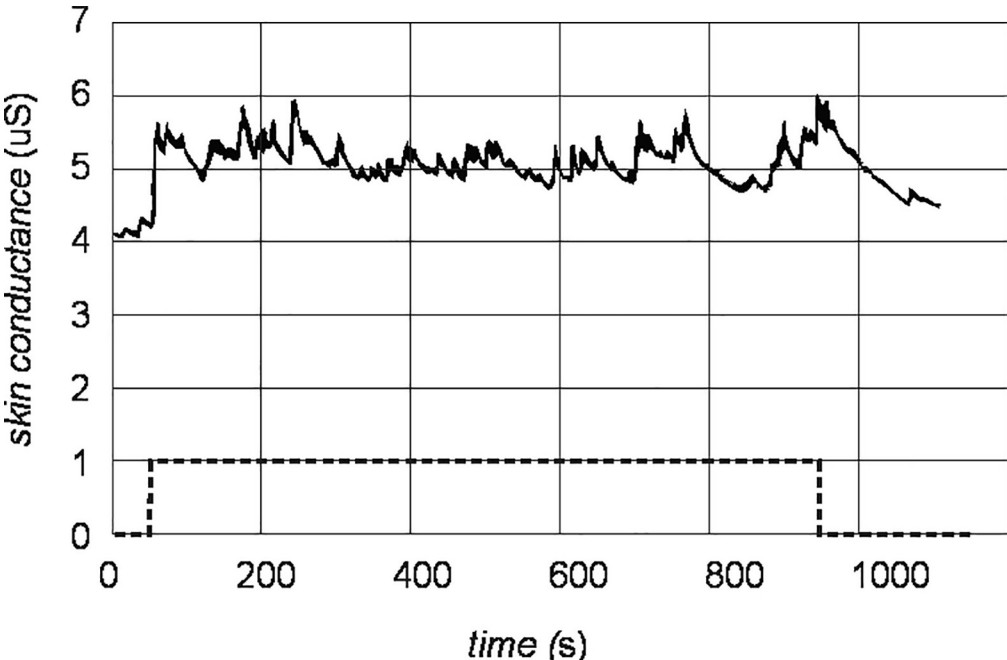

**Fig 3.** Typical raw EDA signal (solid line) of a participant during a mental task (dashed line).

Exosomatic measurements by means of a DC current (Fig 2), represent the basis of today's most widely used instruments. They are predominantly used because of their simplicity, the need for only two electrodes, and possibility of monitoring both tonic and phasic EDA signals. They do, however, lack some advantages of endosomatic method (which is a very unobtrusive method with no special amplifying and coupling systems needed) and AC exosomatic method (no electrode polarization issues) [1,36]. There is a huge variety of EDA devices available (portable, battery powered, embedded or built-in in other settings (e.g. into a computer mouse or steering wheel), with wet or dry electrodes, equipped with only SCL measurement function or additional SCR detection built-in algorithm, logging function etc.).

Sampling frequency of a typical EDA device should be in the order of 10 Hz and above, but this strongly depends on the signal processing the researcher wants to perform. If phasic skin conductance response (SCR) and other fast changing events in electrodermal activity are needed, the sampling rate should be at least 200 Hz, 1 kHz or even 2 kHz being the most common values with the desktop laboratory measuring systems [3,34]. Wearable and especially wireless streaming systems usually have a lower acquisition rate (up to 32 Hz).

Typical acquired raw EDS signal is shown in Fig 3.

## Testing of EDA devices

To ensure the reliable and accurate EDA measurement, one of the simplest tests a researcher should perform before every measurement is static calibration test of the EDA measuring equipment. By connecting the EDA device's electrodes to a fixed resistor (e.g. 1% precision), the resulting skin conductance can be quickly checked in static conditions (Fig 4).

In order to check the functionality and response of the EDA device a simple dynamic test should be performed after the attachment of electrodes to the subject and adequate EDA signal dynamics visually checked by using startle stimuli. Namely, deep breaths, a sudden sound (a loud clap), slap on the inner lower arm or cheek, coughing, scratching, light pinching should

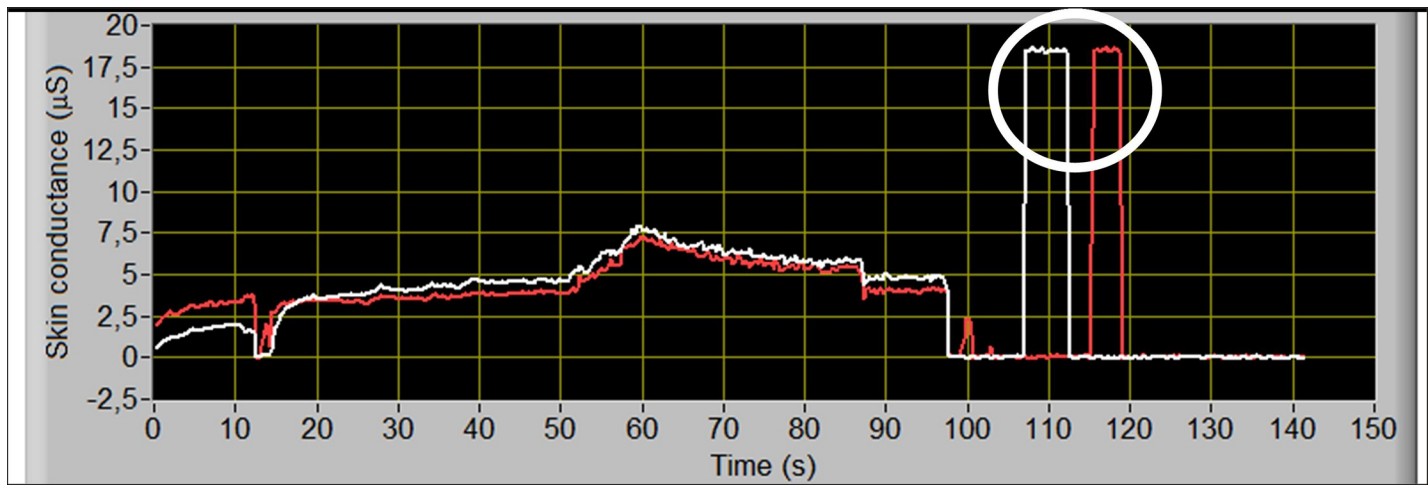

**Fig 4. Raw acquired EDA signal during testing of a dual sensor EDA device (two electrodes acquisition waveforms are represented by the white and red line).** After first 100 seconds EDA device was tested using a fixed resistor of 54 kΩ, corresponding to 18.5 uS (encircled).

result in an increase of the SC level after a second or so. Note, that if such tests fail, even when the EDA device works, the subject could be a non-responder (there are between 5% and 25% of non-responders in normal population) [3,34].

For a more thorough test of the dynamics of EDA instrumentation, we are proposing a concept of a EDA patient simulator, which can be used as a reliable and repeatable device for testing the dynamic functionality and measuring reliability of a EDA device.

## EDA patient simulator

### Requirements of an EDA simulator

Based on EDA phenomena and our experience with skin conductance measuring methods, some possible requirements of an EDA simulator were identified: i) metrologically accurate output, ii) capability of static calibration of EDA devices (e.g. several fixed resistances for measuring range 1 uS to 20 uS), iii) capability of dynamic evaluation of EDA devices (e.g. generation of SCL and SCR, generation of moving artefacts etc.), iv) simple intuitive user interface (no specialized knowledge of EDA phenomena needed), v) portability (battery powered), vi) suitable safety level (medical grade galvanic isolation), vii) a certain degree of automation (e.g. fully automatic simulator with options of manually adjusting certain parameters), viii) possibility of testing EDA devices with different electrode types (e.g. single-use, wet and dry electrodes, shape and type of the electrodes, possibility of device-only testing (without electrodes)).

### EDA simulator design

In this study, a prototype of EDA simulator was designed as a variable resistor, capable of setting the conductance levels of up to 20 uS (i.e. above 50 kΩ) with addition of pulses, corresponding to SCR phasic pulses.

The variable resistor had to fulfil the following specifications: 0 to 500 kΩ range, fast time response, galvanic isolation, and high electrical strength. During the designing and building process several possibilities of variable resistors were checked; voltage-controlled resistor (JFET or MOSFET transistor), temperature-controlled resistor (temperature sensor), force-controlled resistor (force transducer), magnetically-controlled resistor (magnetoresistance),

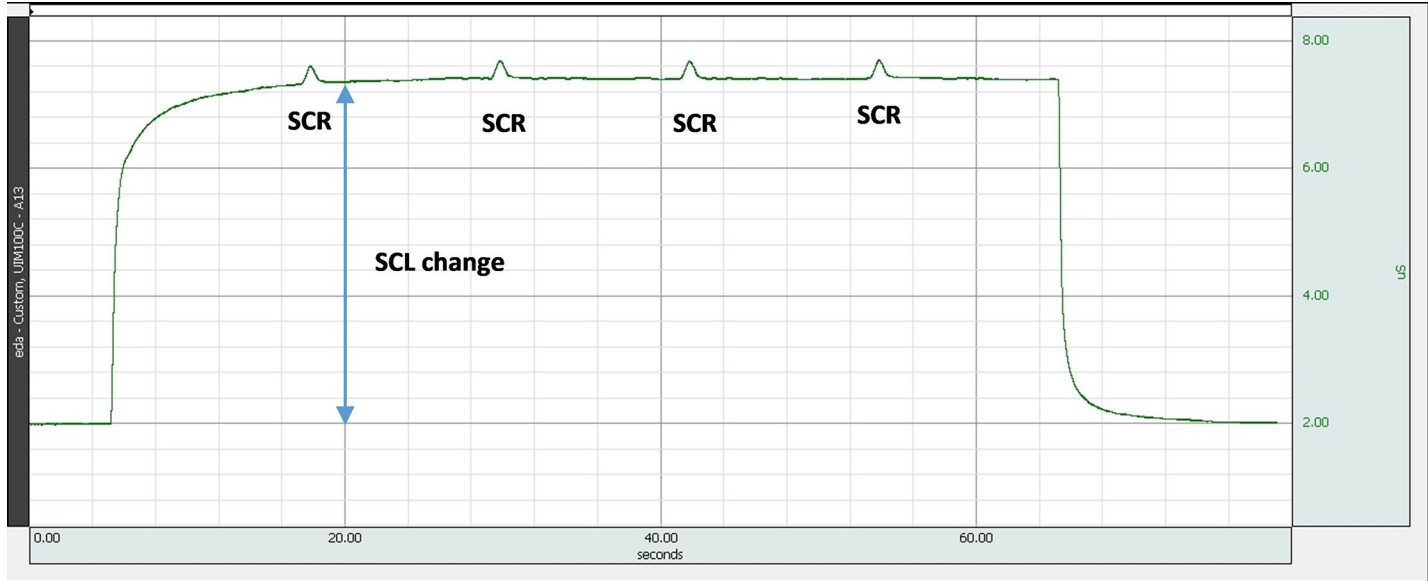

**Fig 5. Skin conductance versus time–an example of an EDA simulator output.** 2 uS electrodermal signal was generated, which increased to 7.2 uS with four superpositioned 0.25 uS SCRs, occurring with 5/min frequency.

programmable resistor (digital potentiometer), and light-controlled resistor (photoresistor, photodiode, optocoupler). Due to an elegant fulfilment of the safety and electrical strength requirement and general simplicity of use, in the final prototype the light-controlled resistance was used. Variable current-controlled resistance was realized by means of an optocoupler—increasing of current through photodiode increased the emitted light and in turn decreased the resistance of the coupled photoresistor (i.e. increased the conductance of the resistor).

In order to check the feasibility and operation of the optocoupler concept, a prototype of EDA simulator was built. The built prototype had an embedded 100 kΩ fixed resistor for static calibration and could generate SCL signals of up to 20 uS and two different SCR amplitudes–small amplitude (0.25 uS) and large amplitude (0.5 uS) (Fig 5). To test EDA device in a wider range of possible psychological arousal levels, the prototype had also a SCR feature—occurrence frequency of SCR could be selected from 5/min, 10/min in 20/min, roughly corresponding to a relaxed, activated and aroused person [1,34].

The main components of the simulator circuit were an LDR optocoupler (NSL-32SR2 by Silonex), an amplification circuit based on LM358AN (by Texas Instruments) and a 32-bit ARM microcontroller developing board (3.3 V Arduino Due SAM3X8E ARM cortex-M3 by Arduino Inc.) (Fig 6). Using these, variable resistor could be set from 0.04 uS to 16 mS (25 MΩ to 60 Ω). The SCL level and SCR amplitudes were set digitally using digital-to-analog output of the Arduino board. Output signal was generated using two-dimensional waveform tables of three EDA signal shapes: SCL signal, small-amplitude SCR signal and high-amplitude SCR signal.

A plastic human hand model was used for EDA device's electrodes attachment. Two fingers were covered with electrically conductive metallized nylon fabric (Shieldex Zell by Statex) plated with silver, copper and tin with surface resistivity of less than 0.02 Ω/ϒ [37] ensuring a reliable electrical contact for the electrodes of the EDA device (Fig 7). Such a set-up is appropriate for testing EDA devices for measuring skin conductance using two electrodes, e.g. on two fingers. On the other hand, a number of different (usually low-cost) EDA devices

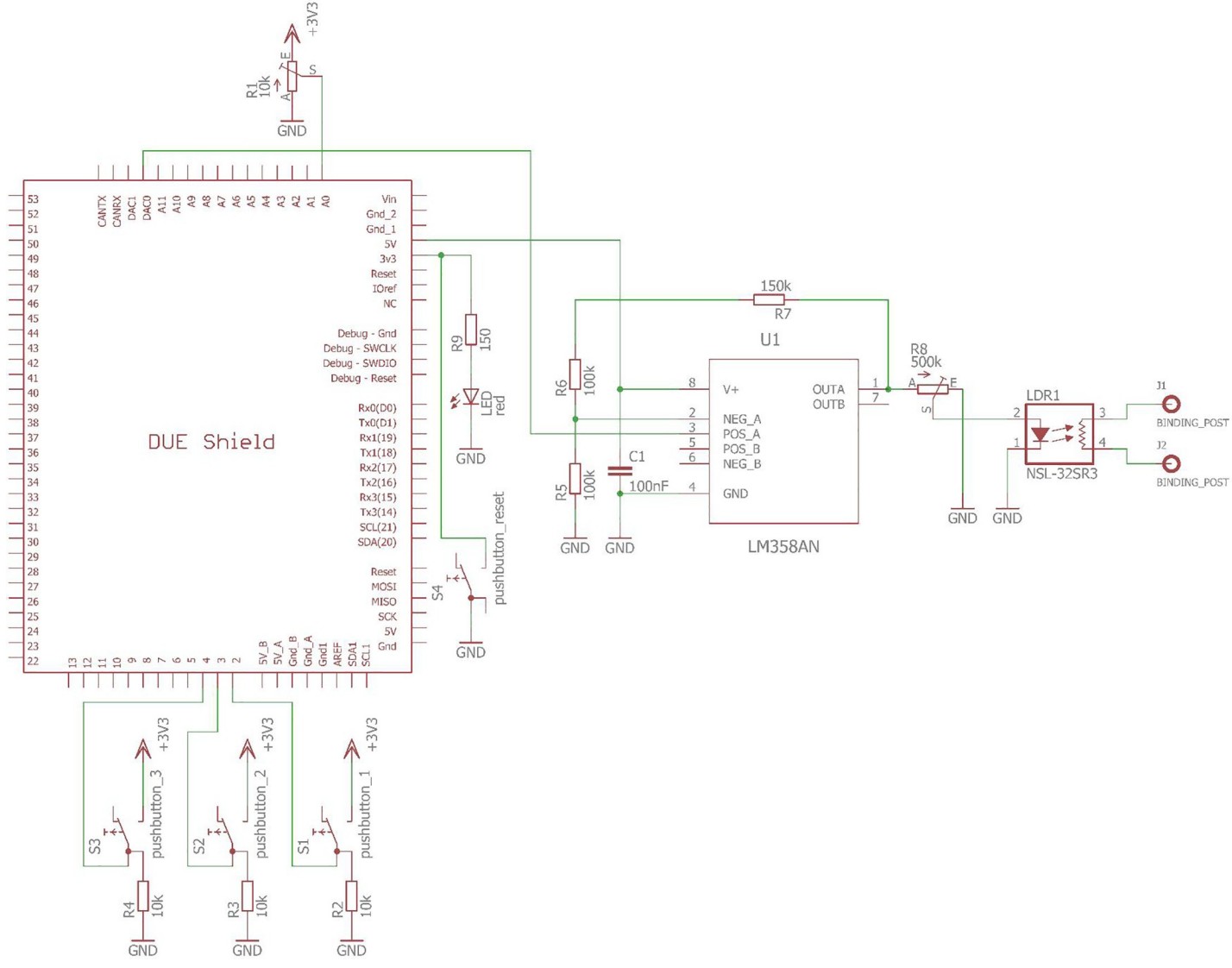

**Fig 6. EDA simulator circuit schematics.**

measuring skin conductance on the user's wrists are used today. To test these, the simulator has built-in EDA terminals, which can be connected to the device's electrodes by simple leads.

One of the basic differences in the waveform of presented EDA simulator in comparison with the real, physiological waveform of human skin is the shape of the time series (Fig 8). Although the simulator waveform looks symmetric and quite artificial, we consider a simplified signal like the one on Fig 8 (right) suitable for the purpose of testing the functionality and basic accuracy of EDA device. For more comprehensive evaluations, the physiology-based signals should be used.

## Testing the functionality of simulator

Using the simulator's built-in precision 100 kΩ resistor (equals 10 uS) a static calibration of a low-cost (under 100 EUR) EDA device was performed (Fig 9). If the fixed resistor itself was calibrated, using calibration by comparison method would allow determination of the

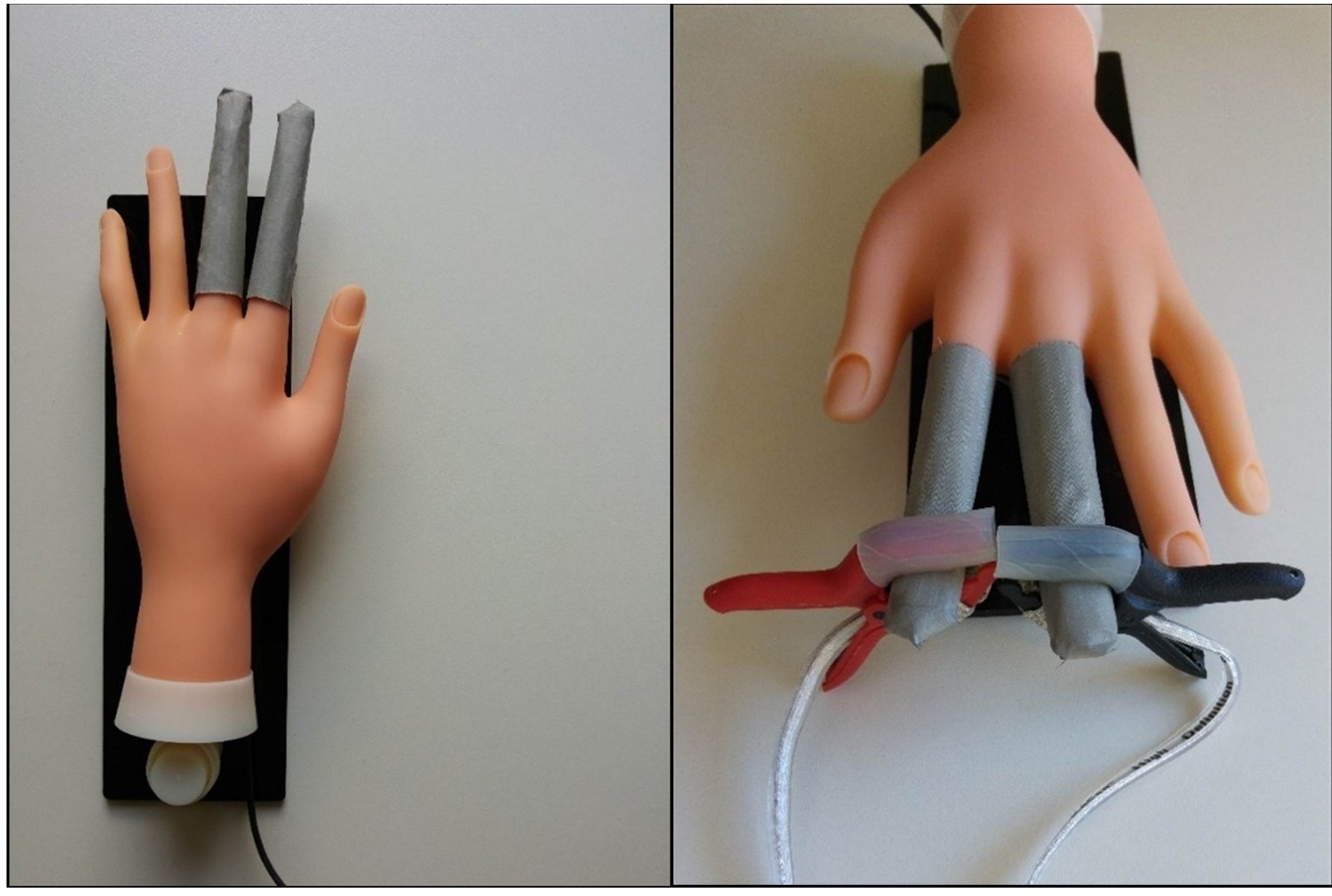

**Fig 7.** (a) EDA simulator with active finger electrodes on index and middle fingers; (b) testing of an EDA device with its clamp electrodes attached to simulator.

measuring error and estimating the uncertainty budget of the EDA device (for details on metrological calibration of EDA devices see [38,39]). After a certain transient time (typically under 0.5 s), time stability of the simulator was estimated by calculating the standard deviation of the output and was in the order of 0.05 uS. Which suffices for the majority of applications using EDA monitoring.

In addition to static calibration, dynamic evaluation of the low-cost EDA device was performed. Reference signal of different amplitudes and different SCR pattern was generated by the simulator (Figs 5 and 10). Maximal dynamic error and uncertainty of an EDA device was calculated to estimate the accuracy of EDA device versus the simulator output. We tested a number of different EDA devices and the errors and uncertainties were below 0.1 uS and 0.3 uS, respectively, error being difference between the EDA reading and simulator setting, and the uncertainty calculated as geometrical sum of simulator's uncertainty, repeatability and reproducibility of the measurement, EDA device repeatability and resolution, resolution of simulator (for details on dynamic evaluations of biomedical instrumentation see [29,38,39]).

In addition to classical (SCL measuring) EDA devices, the simulator's performance was tested also using an EDA device, equipped with an automated SCR detection function. Different frequencies of occurrence of SCR pulses were set on simulator (Fig 10 shows the EDA device acquisition waveform). In addition, EDA device's SCR detection algorithm could be

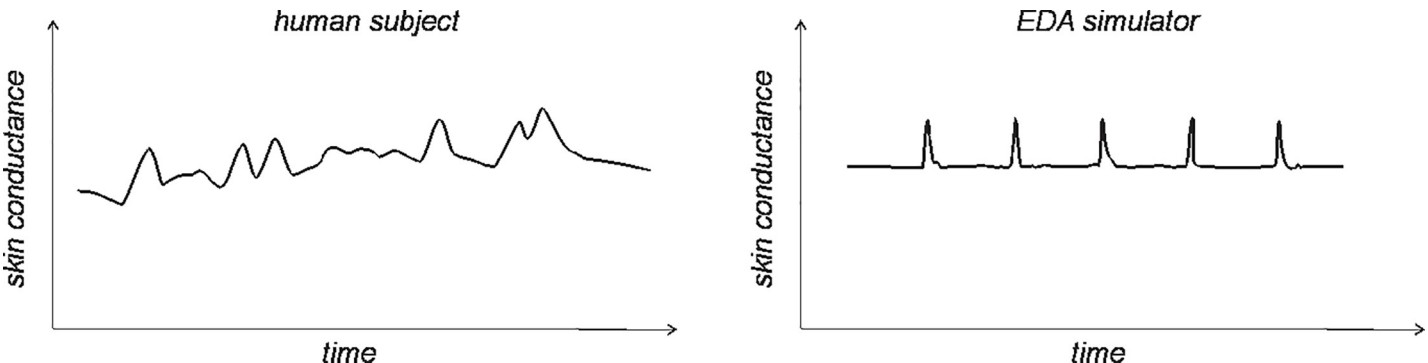

**Fig 8.** Schematics of real physiological waveform (left) and artificially generated output waveform of an EDA simulator (right).

tested. Fig 11 shows raw signal of the EDA device acquired with simulator set to two different amplitudes of the SCR. Tested EDA device's automated SCR detection was set to a 0.3 uS threshold and the water drops indicate the detected SCR (Fig 11).

Table 1 contains the performance of the EDA simulator regarding stability. If we define a target uncertainty for a common EDA measuring device of approx. 0.1 uS, the results show, that the built simulator is adequately stable to reliably check common EDA measuring devices.

## Discussion and conclusions

Measurement of electrodermal activity is a rather simple measuring method using simple and low-cost measuring instruments. It is more and more used in different areas of research, from medicine, ergonomics, biomedical and control engineering, robotics and ergonomics, psychology and education, sports, entertainment to social science and economics. But it is never seriously doubted, in the sense of determining measuring accuracy and quality of measuring result like in [38–40].

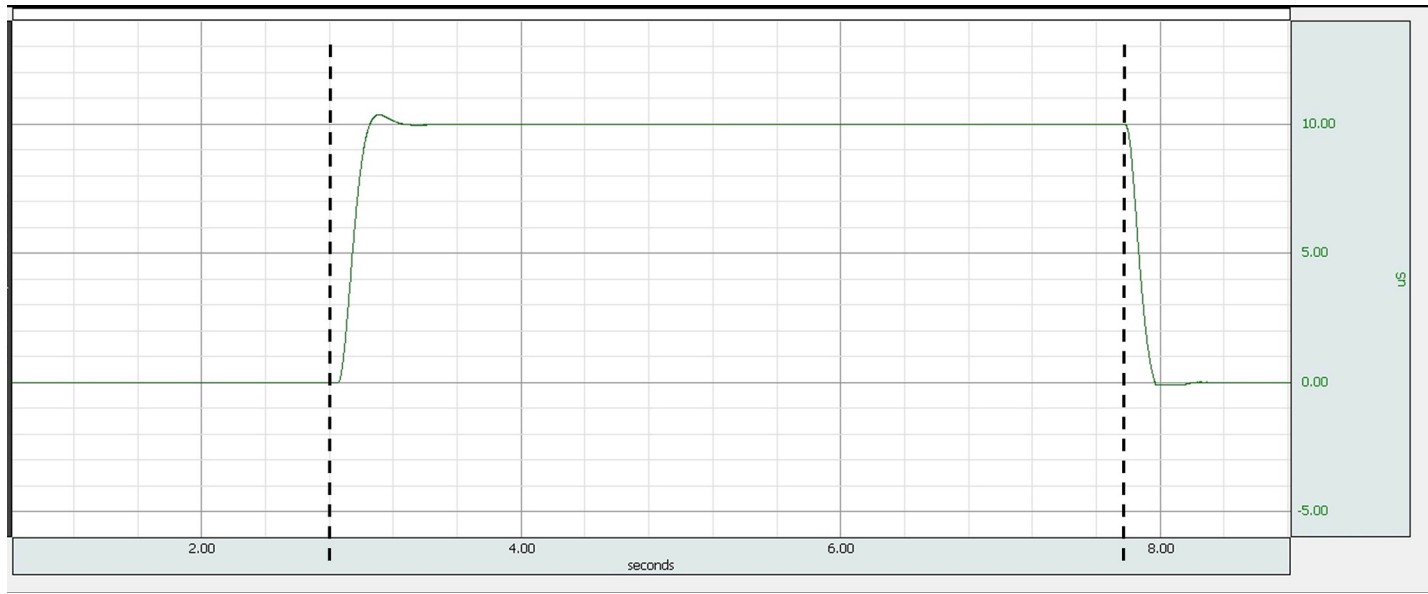

**Fig 9. EDA device acquisition of skin conductance, generated by EDA simulator during static calibration.** In time interval marked with dashed lines, EDA device was measuring a fixed 100 kΩ (10 uS) resistor.

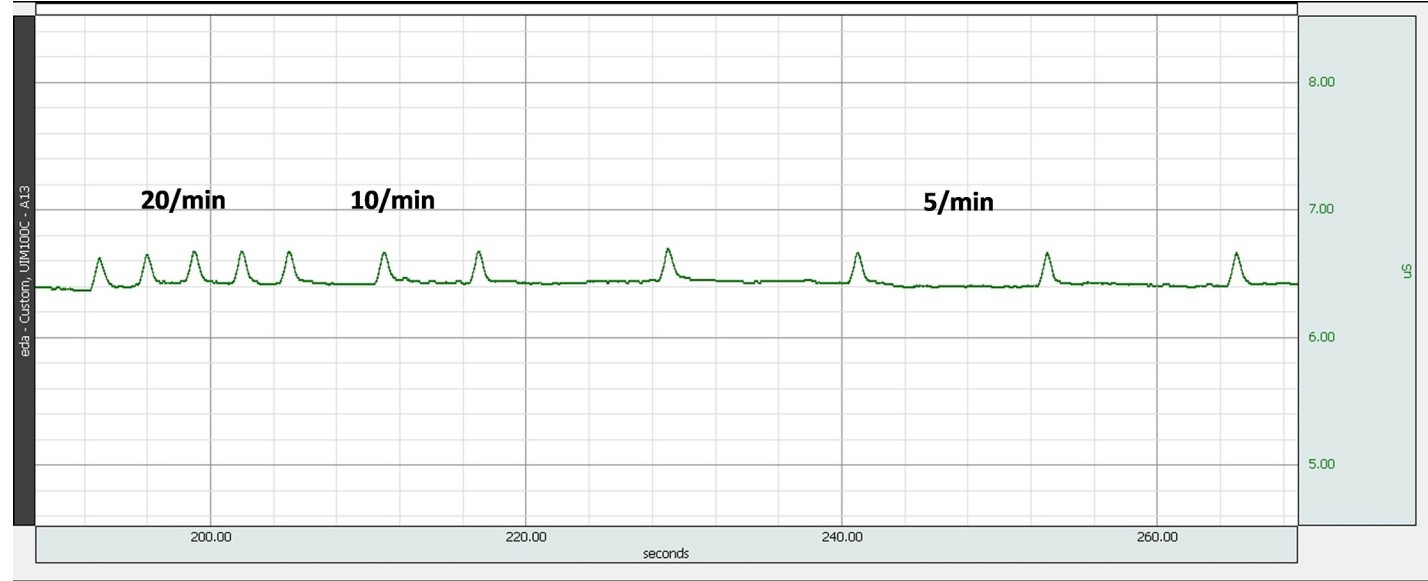

**Fig 10. EDA device acquisition of skin conductance, generated by EDA simulator.** Simulator generated SCRs of different frequencies (5 SCR per min, 10 SCR per min and 20 SCR per min).

In this paper a concept of an EDA simulator for testing EDA devices is, to the best of our knowledge, presented for the first time. EDA simulator is a device capable of generating artificial waveforms of electrical resistance, in part similar to physiological ones. The simulator's main purpose is evaluating the functionality and measuring quality of an EDA device. We presented the simulator's design and basic functionality. Different levels of static EDA signal can be set (i.e. different SCL level). In addition, SCR pulses can be generated, varying in amplitudes and occurring frequency.

While the embedded optocoupler is an elegant solution for constructing a variable resistor, at the same time it is also a major limitation of the proposed EDA simulator. Optocouplers are

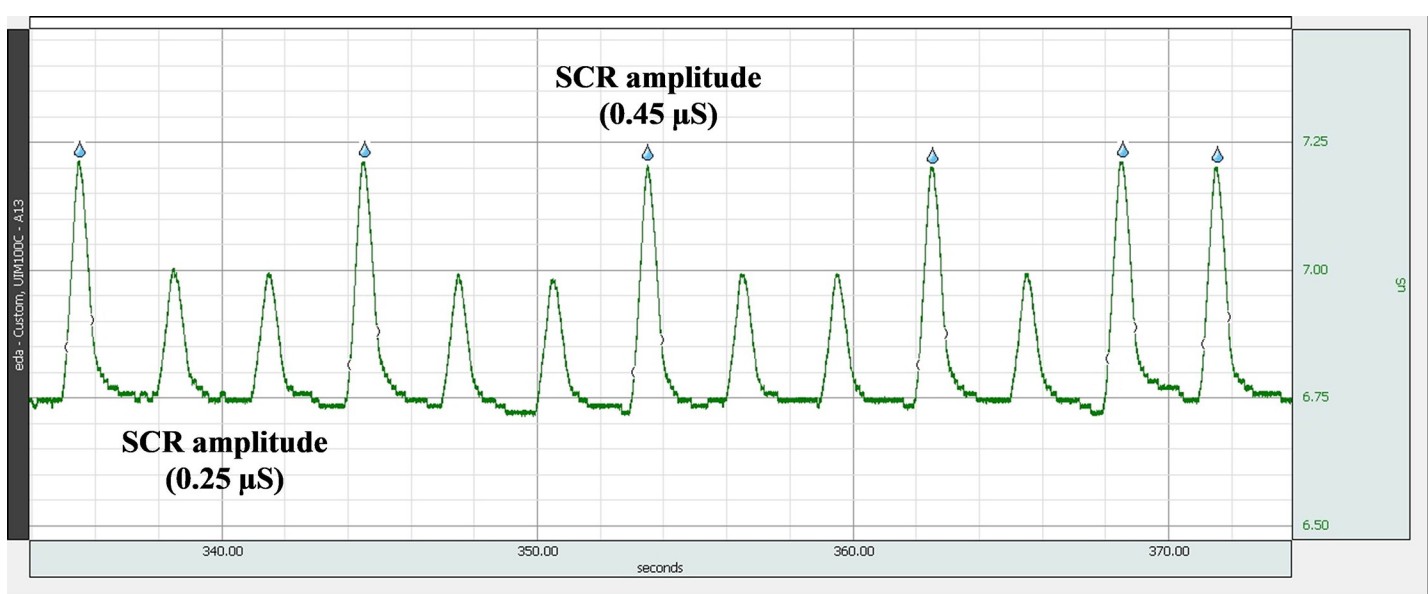

**Fig 11. EDA device acquisition of skin conductance, generated by EDA simulator.** Simulator generated SCRs of different amplitudes (0.25 uS and 0.45 uS).

**Table 1. Stability of the EDA simulator output (all values in uS and measured within a 10 min interval).**

| Simulator settings | Average value | Standard deviation |
|---|---|---|
| frequency 5/min | 4.93 | 0.091 |
| frequency 10/min | 9.94 | 0.082 |
| frequency 20/min | 19.96 | 0.080 |
| SCL (fixed resistor 10 uS) | 9.99 | 0.001 |
| SCL (fixed resistor 4.5 uS) | 4.53 | 0.001 |
| small SCR amplitude | 0.26 | 0.005 |
| large SCR amplitude | 0.45 | 0.005 |

electronic elements not intended for precise operation and during our design, the optimal optocoupler was selected from a batch of elements which had consistent resistance values within the range. The inaccuracies resulted in a certain instability of shape and amplitudes of the generated waveform (see pulses in Fig 11). In the process of dynamic evaluation, an additional simulator limitation is originating from the use of optocoupler—latency of up to 100 ms. The time constant is in the order of seconds (Fig 5), therefore test of EDA device should only be performed after a transient time. Another error source could in principle be the optocouplers temperature dependence, which in our experiments was controlled by maintaining stable laboratory environmental conditions.

EDA simulators main objective is to determine EDA device's accuracy and functionality (e.g. checking its algorithm for SCR detection). Using tests of different EDA devices (with dry stainless steel and gold electrodes, clamp dry electrodes, wet Ag/AgCl electrodes, continuous and intermittent measurements), the built EDA simulator was proven to represent a reliable apparatus for quick tests and even thorough evaluations of any EDA device. Nevertheless, for reliable electrodermal activity measurements, in addition to tests using the EDA simulator, repeatability of an EDA devices should still be additionally evaluated using measurements on a human subject (instructing actions like deep breaths, coughing, scratching, pinching which should result in an increase of the SCL).

A very useful feature of the EDA simulator would be capability of testing several devices simultaneously. This would allow for a fast check of numerous devices (e.g. for experiments in education, where around 30 devices are worn by the pupils and they all need to be checked periodically [41]). The current simulator design is not capable of providing stable resistance values for two or more EDA devices, since the current they are feeding to the simulator's fingers interfered with each other measuring function and the simulator itself. The future versions of simulator design are planned to include this feature.

It has to be noted that the generated signal has an artificial, non-physiological shape, i.e. it is only an approximation of real physiological electrodermal signal. Which in fact is true also for the majority of other patient simulators (e.g. BP, oximetry) [29,30,32]. In the future, real physiological waveforms could be acquired for a more genuine depiction of the dynamics of the human skin. In principle, the future simulator should be able to generate a wider range of EDA behaviours, e.g. moving artefacts, complex changes of both SCL and simultaneous SCR.

## Author Contributions

**Conceptualization:** Gregor Geršak, Janko Drnovšek.

**Investigation:** Gregor Geršak.

**Methodology:** Gregor Geršak, Janko Drnovšek.

**Project administration:** Gregor Geršak.

**Visualization:** Gregor Geršak.

**Writing – original draft:** Gregor Geršak, Janko Drnovšek.

**Writing – review & editing:** Gregor Geršak, Janko Drnovšek.

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
