## [Decision Letter · Decision Letter 0]

22 Nov 2019

PONE-D-19-26536

Electrodermal activity patient simulator

PLOS ONE

Dear Dr Geršak,

Thank you for submitting your manuscript to PLOS ONE. After careful consideration, we feel that it has merit but does not fully meet PLOS ONE’s publication criteria as it currently stands. Therefore, we invite you to submit a revised version of the manuscript that addresses the points raised during the review process.

We would appreciate receiving your revised manuscript by Jan 06 2020 11:59PM. To enhance the reproducibility of your results, we recommend that if applicable you deposit your laboratory protocols in protocols.io, where a protocol can be assigned its own identifier (DOI) such that it can be cited independently in the future. For instructions see: http://journals.plos.org/plosone/s/submission-guidelines#loc-laboratory-protocols

We look forward to receiving your revised manuscript.

Kind regards,

Dominic Micklewright, PhD CPsychol PFHEA FBASES FACSM

Academic Editor

PLOS ONE

1. Please ensure that you refer to Figure 9 in your text as, if accepted, production will need this reference to link the reader to the figure.

Reviewers' comments:

Reviewer's Responses to Questions

**Comments to the Author**

1. Is the manuscript technically sound, and do the data support the conclusions?

Reviewer #1: Partly

Reviewer #2: Partly

2. Has the statistical analysis been performed appropriately and rigorously? 

Reviewer #1: No

Reviewer #2: No

3. Have the authors made all data underlying the findings in their manuscript fully available?

Reviewer #1: No

Reviewer #2: Yes

4. Is the manuscript presented in an intelligible fashion and written in standard English?

Reviewer #1: Yes

Reviewer #2: Yes

5. Review Comments to the Author

Reviewer #1: The authors present a device to simulate electrodermal activity (EDA) data, which is widely used in psychology, neuroscience, economics, and other fields of scientific inquiry. The EDA simulator is timely given the recent increase in the number of EDA recording devices, which often have unknown performance and quality and are often not subject to independent validation studies. The authors describe development of an electrical circuit to simulate the EDA signal and they provide examples of recorded data. They stress that the EDA simulator can only supplement existing tests of EDA device performance on live human subjects, and is not a replacement for human subjects research.

Overall, I am very excited about this manuscript. I think it is interesting and timely and could be very useful for many researchers using EDA. My questions and concerns relate to the device’s precision and how researchers can use this in the real world. Is this device for sale? Or are the authors providing instructions for how other researchers can build this?

Specific major concerns/questions:

• I would like to see more clear metrics (e.g., a table) of the stability of this EDA simulator device: what is the precision in uS and in seconds for the device to deliver what should be an identical time course of SCL vs. Time. For example, if you program the device to deliver a stable baseline for 1 min, followed by 5 SCRs/min for 1 min, then 10 SCRs/min for 1 min, then 20 SCRs/min for 1 min, then another stable baseline for 1 min…and then repeat that 10 times, how consistently does it deliver that time course? How consistently does it deliver the SCRs in time? And how consistent are the SCR amplitudes? I am hoping the device is much more precise that we would require of an EDA measurement device, otherwise it would not be possible to test whether the EDA measurement device performs sufficiently.

• Is there a difference in measurement depending on where you place the EDA electrodes on the EDA simulator’s fingers? Or is the conductance uniform throughout? Is this easy for a user to figure out where to place the electrodes?

• There is a growing popularity in wrist-based EDA recording devices, and these are precisely the devices that are in need of being tested for validation. By comparison, most of the EDA devices that can measure from the finger tips are research-grade devices which are presumably less suspect and less in need of testing for validation. Would the proposed finger-based EDA simulator be able to work with those devices as well? Perhaps a wrist-based EDA simulator can be prepared or this can be suggested for future work?

• Correct me if this is wrong, but I envision using this device with multiple candidate EDA recording devices connected simultaneously to the EDA simulator and then comparing the recorded data from those devices. I would compare each recorded dataset to that generated from the EDA simulator and then compare the devices to each other. Is that correct? If so, how many EDA recording devices can be used at once? Will this interfere with the simulations? If not, then it is even more important to quantify the limits of the EDA simulator (per my first question/concern) because the EDA devices would then have to be tested one at a time, presumably against a standard simulated EDA time course.

Other/minor questions and concerns

• Introduction

• In general, please avoid suggesting that there is a one to one mapping between EDA (or any physiological response) and psychological state, as there is no evidence for this except possibly for very constrained and artificial experimental contexts. As the authors point out there are many other factors that also influence EDA, and there are many other factors that also influence psychological states independent of EDA. For more information, see https://psycnet.apa.org/record/2014-10678-012

• Examples in the paper (not an exhaustive list)—“When a person is psychological aroused, excited or activated, his/her EDA signal increases”

• I understand this is a somewhat theoretical point that is not central to the development of an EDA simulator, but nevertheless I think the research on EDA should be described more accurately

• Please define ICT

• 1-2 kHz seems way oversampled for EDA—I am not aware of any studies using 2 kHz. Often EDA data are downsampled to something more manageable such as 64 Hz or 128 Hz without significant loss of information, because the signal changes in SCRs and SCL are relatively slow

• Fig 10—seems like the labels for 5/min and 20/min are swapped

Discussion

• Several other papers acknowledge the importance of the measuring EDA and the device quality, please also cite, e.g., https://psyarxiv.com/a9ju4/, https://ieeexplore.ieee.org/document/7508621,

• The authors write—“Using extensive tests of different EDA devices (dry stainless steel and gold electrodes, clamp dry electrodes, wet Ag/AgCl electrodes, continuous and intermittent measurements) the built EDA simulator was proven to represent a reliable apparatus for quick tests and even thorough evaluations of any EDA device”—but readers do not get to see any of these data systematically. I would like to see data from the EDA simulator plus the recorded values from the EDA device overlaid or in subplots.

Reviewer #2: The authors present an electrodermal activity simulator that can be used to calibrate a wide variety of standard electrodermal recording devices. In general, I believe that the concept is sound, that the developed device is likely to be effective, and that a somewhat appropriate evaluation has been carried out. However, I have multiple concerns with the work, and thus recommend a major revision. Specific comments below.

MAJOR ISSUES

1. The evaluation of the developed device (section 3) is described in a haphazard fashion, with no systematic description of the protocol or even a description of specific EDA devices. The authors state that they performed static and dynamic evaluation of "an EDA device", but it is unclear what device was used. They then state that "they tested a number of different EDA devices and the errors and uncertainties were below 0.1 us and 0.3 us". However, they again do not state what these "different EDA devices" were, and do not give specific results. The same issue occurs slightly later (line 276), where the authors mention an EDA device with an automated SCR detection algorithm, but again do not state what device was used.

2. In a critical statement (lines 239-241), authors state that "a simplified signal like the one on Fig. 8 was proven suitable for the purpose of testing", but there is no evidence to back up this statement. This is a major issue, as the authors repeatedly acknowledge the weaknesses of the generated waveforms (which look very nonphysiological), but appear to dismiss the concern with this unsupported statement.

3. It is unclear why light-controlled resistance was used even though this is a major component of the work. On line 197, authors state "elegant fulfillment of the safety and electrical strength requirement and general simplicity of use", but it is not clear (at least to me) why a simple voltage-controlled resistor could not be used more easily. Please clarify in more detail.

4. The abstract is, in my opinion, inappropriate. Over half of it is just a background description, and essentially no methods or results are presented.

5. Many statements are not supported by reference. For example, lines 98-101 define SCL and SCR but no reference is given for this definition.

MINOR ISSUES

1. The quality of English could be improved, and I recommend professional editing.

2. Please do not use acronyms without introducing them (e.g., "ICT" on line 67, "EDS" on line 146).

3. Please do not use acronyms in figure captions unless they are defined in the caption itself (e.g., "EDA" in Fig. 3).

4. Line 219: Authors say "SCL level", which would expand to "skin conductance level level".

5. The authors do not clearly state whether any other EDA patient simulators exist.

6. PLOS authors have the option to publish the peer review history of their article (what does this mean?). If published, this will include your full peer review and any attached files.

Reviewer #1: No

Reviewer #2: No

---

## [Author Response · Author response to Decision Letter 0]

8 Jan 2020

PONE-D-19-26536

Electrodermal activity patient simulator

PLOS ONE

Dear Prof Micklewright,

Hereby I am attaching our responses to reviewers’ comments and concerns. The rest of this document is colour coded as following: Black text – reviewers’ comments, Blue text – author’s reply, Red text – changes in the paper.

Thank you for your and the reviewers’ valuable comments and thank you for your consideration.

Best wishes,

Gregor Geršak

Dear Dr Geršak,

Thank you for submitting your manuscript to PLOS ONE. After careful consideration, we feel that it has merit but does not fully meet PLOS ONE’s publication criteria as it currently stands. Therefore, we invite you to submit a revised version of the manuscript that addresses the points raised during the review process.

We would appreciate receiving your revised manuscript by Jan 06 2020 11:59PM. To enhance the reproducibility of your results, we recommend that if applicable you deposit your laboratory protocols in protocols.io, where a protocol can be assigned its own identifier (DOI) such that it can be cited independently in the future. For instructions see: http://journals.plos.org/plosone/s/submission-guidelines#loc-laboratory-protocols

• A rebuttal letter that responds to each point raised by the academic editor and reviewer(s). This letter should be uploaded as separate file and labeled 'Response to Reviewers'.

• A marked-up copy of your manuscript that highlights changes made to the original version. This file should be uploaded as separate file and labeled 'Revised Manuscript with Track Changes'.

• An unmarked version of your revised paper without tracked changes. This file should be uploaded as separate file and labeled 'Manuscript'.

We look forward to receiving your revised manuscript.

Kind regards,

Dominic Micklewright, PhD CPsychol PFHEA FBASES FACSM

Academic Editor

PLOS ONE

Editor's comment:

1. Please ensure that you refer to Figure 9 in your text as, if accepted, production will need this reference to link the reader to the figure.

Thank you for your comment. The text was corrected accordingly and references added.

Using the simulator’s built-in precision 100 kΩ resistor (equals 10 uS) a static calibration of EDA device was performed (Fig 9).

Reviewers' comments:

Reviewer #1: The authors present a device to simulate electrodermal activity (EDA) data, which is widely used in psychology, neuroscience, economics, and other fields of scientific inquiry. The EDA simulator is timely given the recent increase in the number of EDA recording devices, which often have unknown performance and quality and are often not subject to independent validation studies. The authors describe development of an electrical circuit to simulate the EDA signal and they provide examples of recorded data. They stress that the EDA simulator can only supplement existing tests of EDA device performance on live human subjects, and is not a replacement for human subjects research.

Overall, I am very excited about this manuscript. I think it is interesting and timely and could be very useful for many researchers using EDA. My questions and concerns relate to the device’s precision and how researchers can use this in the real world. Is this device for sale? Or are the authors providing instructions for how other researchers can build this?

Using the EDA for a number of years and at the same time being a metrology laboratory, concerned about terms such as accuracy, error and uncertainty we are well aware of the practical limitations of nowadays EDA measuring devices. The goal of our paper was to present a concept of patient simulator and explain one of possible solutions (i.e. instructions for building a simulator).

Specific major concerns/questions:

• I would like to see more clear metrics (e.g., a table) of the stability of this EDA simulator device: what is the precision in uS and in seconds for the device to deliver what should be an identical time course of SCL vs. Time. For example, if you program the device to deliver a stable baseline for 1 min, followed by 5 SCRs/min for 1 min, then 10 SCRs/min for 1 min, then 20 SCRs/min for 1 min, then another stable baseline for 1 min…and then repeat that 10 times, how consistently does it deliver that time course? How consistently does it deliver the SCRs in time? And how consistent are the SCR amplitudes? I am hoping the device is much more precise that we would require of an EDA measurement device, otherwise it would not be possible to test whether the EDA measurement device performs sufficiently.

Thank you for your comment. We performed additional tests of the stability of the level of SC within static tests (with two fixed resistors), amplitude and frequency of pre-set SCR peaks. We included an additional table with data describing the stability of the built simulator. 

Table 1 contains the performance of the EDA simulator regarding stability. If we define a target uncertainty for a common EDA measuring device of approx. 0.1 uS, the results show, that the built simulator is adequately stable to reliably check common EDA measuring devices.

Table 1. Stability of the EDA simulator output (all values in uS and measured within a 10 min interval).

Simulator settings Average value Standard deviation

frequency 5/min 4.93 0.091

frequency 10/min 9.94 0.082

frequency 20/min 19.96 0.080

SCL (fixed resistor 10 uS) 9.99 0.001

SCL (fixed resistor 4.5 uS) 4.53 0.001

small SCR amplitude (uS) 0.26 0.005

large SCR amplitude (uS) 0.45 0.005

• Is there a difference in measurement depending on where you place the EDA electrodes on the EDA simulator’s fingers? Or is the conductance uniform throughout? Is this easy for a user to figure out where to place the electrodes?

We performed some additional measurements to find the sensitivity of the measured EDA value versus the electrode position on the simulator’s fingers. The differences in values (at constant SC conditions) were under 1 uS in the time span of several minutes (i.e. 10 min).

• There is a growing popularity in wrist-based EDA recording devices, and these are precisely the devices that are in need of being tested for validation. By comparison, most of the EDA devices that can measure from the finger tips are research-grade devices which are presumably less suspect and less in need of testing for validation. Would the proposed finger-based EDA simulator be able to work with those devices as well? Perhaps a wrist-based EDA simulator can be prepared or this can be suggested for future work?

Thank you for your comment and a valuable observation. Indeed, the majority of “suspicious” EDA devices are wrist devices. In the first version of the paper, we stated “One of the identified physical limitations is also inability to test wrist-worn EDA devices due to the present configuration of the simulator (designed for finger electrodes only).”, but meanwhile we realised this and corrected the text. The simulator actually does offer a possibility of testing wrist-worn devices, because it has built-in multipurpose output terminals (connectors), which can be connected to the device’s embedded electrodes by simple leads. 

We added additional explanation.

Such a set-up is appropriate for testing EDA devices for measuring skin conductance using two electrodes, e.g. on two fingers. On the other hand, a number of different (usually low-cost) EDA devices measuring skin conductance on the user’s wrists are used today. To test these, the simulator has built-in EDA terminals, which can be connected to the device’s electrodes by simple leads.

• Correct me if this is wrong, but I envision using this device with multiple candidate EDA recording devices connected simultaneously to the EDA simulator and then comparing the recorded data from those devices. I would compare each recorded dataset to that generated from the EDA simulator and then compare the devices to each other. Is that correct? If so, how many EDA recording devices can be used at once? Will this interfere with the simulations? If not, then it is even more important to quantify the limits of the EDA simulator (per my first question/concern) because the EDA devices would then have to be tested one at a time, presumably against a standard simulated EDA time course.

Thank you for your comment. You are absolutely right. If the simulator would be capable of testing several devices simultaneously this would indeed be a very useful feature for fast check of numerous devices (e.g. for experiments in education, where around 30 devices are worn by the pupils and they all need a check – see reference below). In fact, our simulator was not capable of providing stable resistance values for two or more EDA devices, since the current they are feeding to the simulator’s fingers interfered with each other and partly with the simulators circuit. [1]). Appropriate text was added.

1. Geršak, V.; Smrtnik, H.; Prosen, S.; Starc, G.; Humar, I.; Geršak, G. Use of wearable devices to study activity of children in classroom ; Case study — Learning geometry using movement. Comput. Commun. 2020, 150, 581–588.

A very useful feature of the EDA simulator would be capability of testing several devices simultaneously. This would allow for a fast check of numerous devices (e.g. for experiments in education, where around 30 devices are worn by the pupils and they all need to be checked periodically). The current simulator design is not capable of providing stable resistance values for two or more EDA devices, since the current they are feeding to the simulator’s fingers interfered with each other measuring function and the simulator itself.

Other/minor questions and concerns

• Introduction

• In general, please avoid suggesting that there is a one to one mapping between EDA (or any physiological response) and psychological state, as there is no evidence for this except possibly for very constrained and artificial experimental contexts. As the authors point out there are many other factors that also influence EDA, and there are many other factors that also influence psychological states independent of EDA. For more information, see https://psycnet.apa.org/record/2014-10678-012

• Examples in the paper (not an exhaustive list)—“When a person is psychological aroused, excited or activated, his/her EDA signal increases”

• I understand this is a somewhat theoretical point that is not central to the development of an EDA simulator, but nevertheless I think the research on EDA should be described more accurately

Thank you for the comment. The reference was added.

Nowadays, the most common assumption is that when a person is psychologically aroused, excited or activated, hers/his EDA signal increases [1–3], although there are also other opinions [4].

• Please define ICT

ICT stands for information and communications technology. The explanation was added.

EDA was used also in ICT (information and communications technology) and entertainment [20–24], education [25,26] and food industry research [27,28].

• 1-2 kHz seems way oversampled for EDA—I am not aware of any studies using 2 kHz. Often EDA data are downsampled to something more manageable such as 64 Hz or 128 Hz without significant loss of information, because the signal changes in SCRs and SCL are relatively slow

We do not agree. Although the skin conductance is a relatively slow physiological signal, the post-festum signal processing is not entirely immune to the sampling rate. In out experiments we use 1 kHz (which might be (too) high, but nowadays research devices have no problem with it)), because we realised that SCR content (the number of SCR) could be compromised if under-sampled (e.g. some wearables offer maximal 32 Hz sampling rate which is not always adequate for a precise and robust SCR analysis). Similar suggestions on sampling frequency can be found also in Figner et al. Using skin conductance in judgment and decision making research, 2011 and in Braithwaite et al. A Guide for Analysing Electrodermal Activity (EDA) & Skin Conductance Responses (SCRs) for Psychological Experiments, 2013.

• Fig 10—seems like the labels for 5/min and 20/min are swapped

Thank you for the comment. The figure was corrected.

Discussion

• Several other papers acknowledge the importance of the measuring EDA and the device quality, please also cite, e.g., https://psyarxiv.com/a9ju4/, https://ieeexplore.ieee.org/document/7508621, • The authors write—“Using extensive tests of different EDA devices (dry stainless steel and gold electrodes, clamp dry electrodes, wet Ag/AgCl electrodes, continuous and intermittent measurements) the built EDA simulator was proven to represent a reliable apparatus for quick tests and even thorough evaluations of any EDA device”—but readers do not get to see any of these data systematically. I would like to see data from the EDA simulator plus the recorded values from the EDA device overlaid or in subplots.

Thank you for the interesting reference. The text was changed - the reference added.

But it is never seriously doubted, in the sense of determining measuring accuracy and quality of measuring result like in [38–40].

Using the built EDA simulator for testing different types of EDA devices (low-cost and research-grade, with dry stainless-steel and golden electrodes, clamp dry electrodes, wet Ag/AgCl electrodes, continuous and intermittent, desktop and portable), it can be concluded that the simulator represents a reliable apparatus for quick tests of EDA devices.

Reviewer #2: The authors present an electrodermal activity simulator that can be used to calibrate a wide variety of standard electrodermal recording devices. In general, I believe that the concept is sound, that the developed device is likely to be effective, and that a somewhat appropriate evaluation has been carried out. However, I have multiple concerns with the work, and thus recommend a major revision. Specific comments below.

MAJOR ISSUES

1. The evaluation of the developed device (section 3) is described in a haphazard fashion, with no systematic description of the protocol or even a description of specific EDA devices. The authors state that they performed static and dynamic evaluation of "an EDA device", but it is unclear what device was used.

They then state that "they tested a number of different EDA devices and the errors and uncertainties were below 0.1 us and 0.3 us". However, they again do not state what these "different EDA devices" were, and do not give specific results. The same issue occurs slightly later (line 276), where the authors mention an EDA device with an automated SCR detection algorithm, but again do not state what device was used.

Thank you for your comment. It is the authors’ opinion that the brand and model name would not give any additional information to the reader (and would represent just an advertising for them). The EDA devices used were a Sensewear device, a Shimmer device, a Biopac device, a g.Tec device and a couple of self-made (and extensively tested and used) devices. We do acknowledge that the device quality is important; therefore, we added the price range for the EDA devices.

Using the simulator’s built-in precision 100 kΩ resistor (equals 10 uS) a static calibration of a low-cost (under 100 EUR) EDA device was performed (Fig 9). 

In addition to static calibration, dynamic evaluation of the low-cost EDA device was performed.

2. In a critical statement (lines 239-241), authors state that "a simplified signal like the one on Fig. 8 was proven suitable for the purpose of testing", but there is no evidence to back up this statement. This is a major issue, as the authors repeatedly acknowledge the weaknesses of the generated waveforms (which look very nonphysiological), but appear to dismiss the concern with this unsupported statement.

The reviewer is right. Our initial statement was very strong. Namely, the fact is, we only performed a limited number of checks within an EDA device test. We changed the text and the statement is now written in a weaker fashion. The reason comes from our experience (we actually did perform a number of tests of very different EDA devices), which in part come from the research area of blood pressure patient simulators, where the same problems regarding the natural physiological versus simplified artificial output occur. For short and fast checks, the artificial signals were proven adequate. For comprehensive tests they are probably not entirely suitable. But the fact is, nobody has proven this empirically. 

Although the simulator waveform looks symmetric and quite artificial, we consider a simplified signal like the one on Fig 8 (right) suitable for the purpose of testing the functionality and basic accuracy of EDA device. For more comprehensive evaluations, the physiology-based signals should be used.

3. It is unclear why light-controlled resistance was used even though this is a major component of the work. On line 197, authors state "elegant fulfillment of the safety and electrical strength requirement and general simplicity of use", but it is not clear (at least to me) why a simple voltage-controlled resistor could not be used more easily. Please clarify in more detail.

In the course of the simulator development, different types of variable resistors were tested (see line 189-195 on page 11), with the emphasis on digipot, programmable resistor, combination light diode-photoresistor and optocoupler. Using simple voltage-controlled resistors we could not reach the wide resistance range easily (for 1 uS to 30 uS we needed aprox. 33 kOhm do 1 MOhm range with resistance resolution of some 0.05 uS (i.e. 20 Mohm)). Therefore, a combination of perfect galvanic isolation, wide output resistance range, no need for microcontroller, ADC and sampling circuits resulted in choosing the optocoupler. Optocoupler really was the simplest, adequately linear solution.

4. The abstract is, in my opinion, inappropriate. Over half of it is just a background description, and essentially no methods or results are presented.

The abstract has been rewritten.

In this paper, we propose a concept of an EDA patient simulator - a device enabling metrological testing of EDA devices by means of a variable resistance. EDA simulator was designed based on a programmable light-controlled resistor with a wide resistance range, capable of simulating skin conductance levels (SCL) and responses (SCR) and was equipped with an artificial hand. The hand included electrically conductive fingers for attachment of EDA device electrodes. A minimal set of tests for evaluating an EDA device was identified, the simulator’s functionality discussed and some testing results presented.

5. Many statements are not supported by reference. For example, lines 98-101 define SCL and SCR but no reference is given for this definition.

Thanks for the comment. We checked the text and made changes.

Tonic, slowly changing part of the SC signal is named skin conductance level (SCL) [1]. Fast phasic pulses are called electrodermal responses, or skin conductance responses (SCR) [1].

As a rule-of-a-thumb, values of a couple of SCR per minute indicate the subject is in relaxed state (baseline) and values above 20 SCR/min indicate an aroused subject [1,34].

MINOR ISSUES

1. The quality of English could be improved, and I recommend professional editing.

2. Please do not use acronyms without introducing them (e.g., "ICT" on line 67, "EDS" on line 146).

Thanks for the comment. The explanation was added.

EDA was used also in ICT (information and communications technology) and entertainment [20–24], education [25,26] and food industry research [27,28].

Typical acquired raw EDA signal is shown in Fig 3.

3. Please do not use acronyms in figure captions unless they are defined in the caption itself (e.g., "EDA" in Fig. 3).

Thanks for the comment. The text was corrected.

Fig 3. Typical raw skin conductance (or EDA) signal (solid line) of a participant during a mental task (dashed line).

4. Line 219: Authors say "SCL level", which would expand to "skin conductance level level".

Thanks for the comment. The text was corrected.

The SCL and SCR amplitudes were set digitally using digital-to-analog output of the Arduino board.

Different levels of static EDA signal can be set (i.e. different SCL).

5. The authors do not clearly state whether any other EDA patient simulators exist.

We have not come across not even an EDA simulator’s concept, yet alone another EDA simulator. In line 315 on page 18 it was stated “In this paper a concept of an EDA simulator for testing EDA devices is, to the best of our knowledge, presented for the first time.«

In this paper a concept of an EDA simulator for testing EDA devices is, to the best of the authors’ knowledge, presented for the first time.

---

## [Editor Report · Decision Letter 1]

28 Jan 2020

Electrodermal activity patient simulator

PONE-D-19-26536R1

Dear Dr. Geršak,

We are pleased to inform you that your manuscript has been judged scientifically suitable for publication and will be formally accepted for publication once it complies with all outstanding technical requirements.

With kind regards,

Dominic Micklewright, PhD CPsychol PFHEA FBASES FACSM

Academic Editor

PLOS ONE

---

## [Editor Report · Acceptance letter]

29 Jan 2020

PONE-D-19-26536R1 

Electrodermal activity patient simulator 

Dear Dr. Geršak:

I am pleased to inform you that your manuscript has been deemed suitable for publication in PLOS ONE. Congratulations! Your manuscript is now with our production department. 

With kind regards,

on behalf of

Professor Dominic Micklewright 

Academic Editor

PLOS ONE